

# Roles of H$_2$S and NO in regulating the antioxidant system of *Vibrio alginolyticus* under norfloxacin stress

Shuhe Chen[*], Yunsheng Chang[*] and Yu Ding

Fisheries College, Guangdong Provincial Key Laboratory of Pathogenic Biology and Epidemiology for Aquatic Economic Animals and Guangdong Key Laboratory of Control for Diseases of Aquatic Economic Animals, Guangdong Ocean University, Zhanjiang, Guangdong, China

[*] These authors contributed equally to this work.

## ABSTRACT

Antioxidant system is of great importance for organisms to regulate the level of excessive reactive oxygen species (ROS) under the environmental stresses including antibiotics stress. Effects of norfloxacin (NOR) on cystathionine-$\beta$-synthase (CBS), nitric oxide synthase (NOS) and antioxidant enzymes were investigated, and interaction between NO and H$_2$S and their regulation on the antioxidant system of *Vibrio alginolyticus* under NOR were determined as well in the present study. After treated with 2 µg/mL NOR (1/2 MIC), CBS content, H$_2$S and NO contents decreased while H$_2$O$_2$ accumulation and the antioxidant-related genes mRNA level increased. Additionally, the endogenous H$_2$S content in *V. alginolyticus* was increased by the exogenous NO, while H$_2$O$_2$ accumulation and the relative expression level of *SOD* (Superoxide dismutase gene) decreased under exogenous NO or H$_2$S. And the content of endogenous NO and NOS in *V. alginolyticus* increased under the exogenous H$_2$S as well. Taken together, these results showed that anti-oxidative ability in *V. alginolyticus* was respectively enhanced by the gas molecules of H$_2$S and NO under NOR-induced stress, and there may be a crosstalk regulative mechanism between H$_2$S and NO. These results lay a foundation for the research of regulation network of H$_2$S and NO, and provide a hint to synthesize anti-vibrio drugs in the future.

## INTRODUCTION

Under the environmental stresses including antibiotics stress, the excessive reactive oxygen species (ROS) produced by bacteria have deleterious effects on their growth and reproduction. Antioxidant system, comprised of antioxidant enzymes and nonenzymatic antioxidants, is of great significance for bacteria to resist and reduce ROS under the adverse condition (*Ding et al., 2012*). Superoxide dismutase (SOD), glutathione reductase (GR) and catalase (CAT) are the significant members of the antioxidant enzymes. Glutathione (GSH) is an important nonenzymatic antioxidant, and it can be regenerated through reduction of GSSG (oxidized glutathione) catalyzed by NADPH-dependent glutathione reductase (GR) (*Hicks et al., 2007*). Hydrogen sulphide (H$_2$S) has been proved to be a

Corresponding author
Yu Ding, dingy@gdou.edu.cn

gasotransmitter (*Nandi, Ravindran & Kurian, 2018*; *Arif et al., 2020*), and plays a vital role in regulating the antioxidant system by activating SOD, a crucial antioxidase, to reduce ROS and balance the oxidation-reduction level in *Escherichia coli* (MG1655) (*Shatalin et al., 2011*). Another ubiquitous gasotransmitter, nitric oxide (NO), which is proved to induce the tolerance of biological cells under high temperature stress by increasing $H_2S$ content (*Li et al., 2013*), also acts as the mediator of antioxidant system to protect bacteria from the antibiotics stress (*Gusarov et al., 2009*). Although the mechanisms and the detailed network of crosstalk between NO and $H_2S$ remain unclear, some studies have demonstrated that the two gases can act synergistically to keep the cellular redox homeostasis (*Shatalin et al., 2011*; *Li et al., 2013*; *Iqbal et al., 2021*).

*Vibrio alginolyticus* is a gram-negative bacterium, and a marine pathogenic species which can cause diseases of many marine animals with heavy economic loss in aquaculture industry (*Ardiç & Ozyurt, 2004*; *Liu et al., 2004*; *Zhou et al., 2013*). In order to prevent the vibriosis, antibiotics is considered as the most effective and economic method, and has already been used widely, even though it can cause new environmental problems. The bactericidal mechanism varies with the different antibiotics (*Walsh, 2000*), but most of them would increase ROS level in the bacterial cells and induce the cellular death (*Davies et al., 2009*; *Dwyer, Kohanski & Collins, 2009*; *Mols & Abee, 2011*). Therefore, it is necessary and reasonable that *V. alginolyticus* uses its antioxidant system to eliminate the excessive ROS to protect itself from the oxidative damage induced by antibiotics. However, whether $H_2S$ and NO can act synergistically in regulating its antioxidant system in *V. alginolyticus* is still unknown.

In this paper, in order to discover and obtain fundamental information about the role of $H_2S$ and NO on the antioxidant system of *V. alginolyticus* treated with NOR, the content of hydrogen sulfide synthase (cystathionine-β-synthase, CBS) and nitric oxide synthase (NOS), $H_2S$ and NO and hydrogen peroxide ($H_2O_2$), and the relative expression levels of antioxidant-related genes such as superoxide dismutase (SOD), glutathione reductase (GR) and catalase (CAT), were measured under different gas donors and scavengers treatment, which is expected to provide a hint to synthesize drugs to resist *Vibrio* spp. in the future.

## MATERIALS AND METHODS

### Bacteria strain, culture media and harvest

*V. alginolyticus* HY9901 strain used in this work was isolated and preserved in our laboratory, Guangdong Provincial Key Laboratory of Pathogenic Biology and Epidemiology for Aquatic Economic Animals in China (*Cai et al., 2007*). Bacteria were cultured in Tryptone Soy Broth medium (TSB with an adjusted pH of 7) with 15 g/L tryptone, 5 g/L soy peptone and 5 g/L NaCl. Bacteria cells were collected by centrifugation. Subsequently, the pellets were resuspended and washed with phosphate buffer saline (PBS) for 2–3 times for the following biochemical measurements after sonication. While RNA was isolated from the collected cells (in 1 mL) without washing after centrifugation.

**Table 1  Primer sequences for RT-PCR.**

| Gene | Sequences (5′- 3′) | Amplicon size (bp) | Reference |
|------|--------------------|--------------------|-----------|
| *SOD* (s) | TTATGGCGTTGTTTTTAC | 162 | This study |
| *SOD* (a) | TGCTTCCCTGTGTTGTTA | | |
| *CAT* (s) | AAAAAGATTGGCAAGGGA | 172 | This study |
| *CAT* (a) | GCGAATGGCACAGATACA | | |
| *GR* (s) | GGTGGTCGTCCTACTATTCC | 196 | This study |
| *GR* (a) | TACGCAGTGGTGACTCTTTAC | | |
| 16S *rDNA* (s) | AAAGCACTTTCAGTCGTGAGGAA | 156 | *Rui et al. (2008)* |
| 16S *rDNA* (a) | TGCGCTTTACGCCCAGTAAT | | |

## Primers and reagents

Genes encoding antioxidant-related enzymes of SOD, GR and CAT in *V. alginolyticus* were cloned and sequenced in our previous work, and qPCR primers were designed according to the sequenced results (Table 1), referenced by 16S rDNA gene. Primers were synthesized by Sangon Biological Engineering Technology & Services Co., Ltd. (China). The kits of RNA extraction and reverse transcription were from Beijing TransGen Biotech Co., Ltd. (China). Sodium hydrosulfide (NaSH, $H_2S$ donor), hypotaurine (HT, $H_2S$ scavenger), sodium nitroprusside (SNP, NO donor) and 2-phenyl-4, 4, 5, and 5-tetramethylimidazoline-1-oxyl 3-oxide (PTIO, NO scavenger) came from Sigma-Aldrich Co. LLC (USA), while NOR came from Guangzhou Technology Company Limited (China). ELISA kits for microorganism CBS and NOS, $H_2S$ and NO were provided by Shanghai Jianglai Industrial Ltd (China). Hydrogen Peroxide Assay Kit and Total Protein Quantification Assay Kit were from Nanjing Jiancheng Bioengineering Institute (China).

## Preparation for stock solutions

NaSH, SNP, HT and PTIO were respectively weighed, and then dissolved and diluted with cold double-distilled water to yield stock solutions in a final concentration of 100.0 mM. Except for NaSH and SNP are prepared before using, all the other liquors were preserved in −20 °C for one week, after filtering through 0.22 μm microfiltration membrane.

0.15 g of NOR was dissolved with 10 mL 12 mol/L cold HCl to yield a stock solution with a final concentration 15,000 μg/mL and preserved in −80 °C for one week.

## Determination of the minimum inhibitory concentration (MIC) of NOR against *V. alginolyticus*

*V. alginolyticus* was respectively cultured in total volume of 0.5 mL at 28 °C for 48 h with NOR from 0.125 to 256 μg/mL (at final concentration) using two-fold serial dilution method. Briefly, the stock solution of NOR was added into 10 mL *V. alginolyticus* culture in logarithmic period with $OD_{600}$ of 0.1, to yield 256 μg/mL NOR treatment, and subsequently 0.5 mL of *V. alginolyticus* culture with 256 μg/mL NOR was transferred into 0.5 mL of *V. alginolyticus* culture in logarithmic period without NOR, to yield 128 μg/mL NOR treatment. Similarly, repeat the above dilution till NOR concentration reach 0.125 μg/mL. As a control, a same volume of HCl was added into 10 mL *V. alginolyticus* culture, and

diluted with the same method as described above. Growth of *V. alginolyticus* in different concentration groups was recorded to determine MIC value of NOR against the bacterium. Every treatment including control were triplicated ($n = 3$).

## Sample treatment with NOR

*V. alginolyticus* was transferred to 50 mL fresh TSB medium and cultured at 28 °C for 10 h till $OD_{600}$ value with 0.5, and subsequently treated with 0 (control) and 1/2 MIC concentration NOR, respectively, and continued to culture for 2 h. Then harvested cells and measured CBS and NOS, $H_2S$ and NO contents, and $H_2O_2$ accumulation as well as the relative expression levels of the antioxidant-related genes. Every treatment was triplicated ($n = 3$).

## Measurement of the total protein content

Total protein was measured according to the bicinchoninic acid method (with standard sample) (*Smith et al., 1985*) with Total Protein Quantification Assay Kit (Nanjing Jiancheng Bioengineering Institute, China). In briefly, the principle of assessment total protein concentration is that $Cu^+$ reduced from $Cu^{2+}$ by proteins in the alkaline condition can react with the bicinchoninic acid (BCA) reagent to form the purple complex which can be spectrophotometrically read at 562 nm, and to quantify the protein by comparing with the standard curve. The absorbance is proportional to the protein concentration, so concentration can be obtained following the formula: total protein = $(OD_{sample}- OD_{blank})/(OD_{standard}- OD_{blank}) \times$ standard sample (524 µg/mL) × sample dilution times.

## Interaction of $H_2S$ and NO

To determine the effect of NO on CBS content and $H_2S$ content, *V. alginolyticus* was treated with a final concentration of the following resolution respectively, (1) control; (2) 1.0 mM SNP; (3) 0.2 mM PTIO under NOR for 2 h. Every treatment was repeated thricely. Similar to the above, in order to determine the effect of $H_2S$ on NOS content and NO content, *V. alginolyticus* was treated with a final concentration of the following resolution respectively, (1) control; (2) 1.0 mM NaSH; (3) 0.2 mM HT for 2 h. Every treatment was triplicated ($n = 3$).

For studying the effect of NO and $H_2S$ on the antioxidant-related enzymes, *V. alginolyticus* was treated for 2 h with a final concentration of the following resolution respectively: (1) control; (2) 1.0 mM SNP; (3) 0.2 mM PTIO; (4) 1.0 mM NaSH; (5) 0.2 mM HT. Every treatment was triplicated ($n = 3$). $H_2O_2$ accumulation level and relative expression level of the antioxidant-related genes were subsequently determined.

## Measurement of CBS and NOS contents

CBS and NOS contents were measured with ELISA method of the quantitative sandwich immunoassay technique (Sandwich ELISA) (*Stynen et al., 1995*). The principles and methods are briefly described that the purified antibody against CBS or NOS was pre-coated in microtiter plate wells in advance, subsequently the sample containing CBS or NOS was added into the microtiter plate, to form antigen-antibody complex through 30 min incubation with closure plate membrane at 37 °C. After washing with wash buffer

and discarding residue liquid by swing completely, 50 μL combined antibody of CBS or NOS with horseradish peroxidase (HRP)-conjugate reagent were introduced into plate wells, forming the antibody-antigen-antibody (HRP enzyme labeled) complex after 30 min incubation at 37 °C. Subsequently, microtiter plate was washed with wash buffer and swing completely, and 50 μL carbamide peroxide [$CO(NH_2)_2 \cdot H_2O_2$)] and 50 μL substrate solution of 3, 3, 5, 5-tetramethylbenzidine (TMB) were added to plate wells. TMB substrate became blue after HRP enzyme catalyzed in 30 min at 37 °C, and this chromogenic reaction was terminated by the addition of 50 μL 2M sulphuric acid solution for the spectrophotometric measurement at 450 nm. The standard curve was set up on the same plate simultaneously, which was used for calculating CBS or NOS contents, and the result was further calibrated by the corresponding total protein concentration. The final results were showed with U/g protein.

## Measurement of H$_2$S and NO contents

H$_2$S contents were measured with the sandwich ELISA method (*Zheng et al., 2016*). Briefly, H$_2$S was caught by the pre-coated antibody in microtiter plate wells in advance, then sample containing H$_2$S were introduced to microtiter plate wells in triplicate, forming antigen-antibody complex after incubation at 37 °C. The following steps refered to the above sandwich ELISA method of CBS and NOS. The result was further calibrated by the corresponding total protein concentration to get the final results (μmol/g protein).

Similar to the sandwich ELISA method of H$_2$S measurement, NO was detected by the ELISA kit according to the manufacturer's instructions (*Abd El Dayem et al., 2019*), and subsequently further calibrated by the corresponding total protein concentration, to yield the final results (μmol/g protein).

## Measurement of H$_2$O$_2$

H$_2$O$_2$ was measured with Hydrogen Peroxide Assay Kit according to the manufacturer's recommendations. The measurement principle was that this kind of ROS can react with molybdic acid to form peroxomolybdic acid complex which can be read at 450 nm. The accumulation of H$_2$O$_2$ in the samples were subsequently recorded by comparing OD value of the samples with the standard.

## Measurement of the relative expression level of antioxidant-related genes

qPCR was used to measure the gene expression level of SOD, GR and CAT. Total RNA was extracted using RNA extraction kit, and subsequently converted to cDNA using the reverse transcription kits. All RNA samples were adjusted to a same concentration prior to the reverse transcription with RNase free water. cDNA was used to qPCR with a program: 1 denaturation cycle at 94 °C for 5 min, 40 amplification cycles at 60 °C for 20 s, and 72 °C for 45 s with Bio-Rad iQ5 Real Time PCR System (USA). 16S rDNA gene of *V. alginolyticus* was used as a reference gene in qPCR. All samples were triplicated.

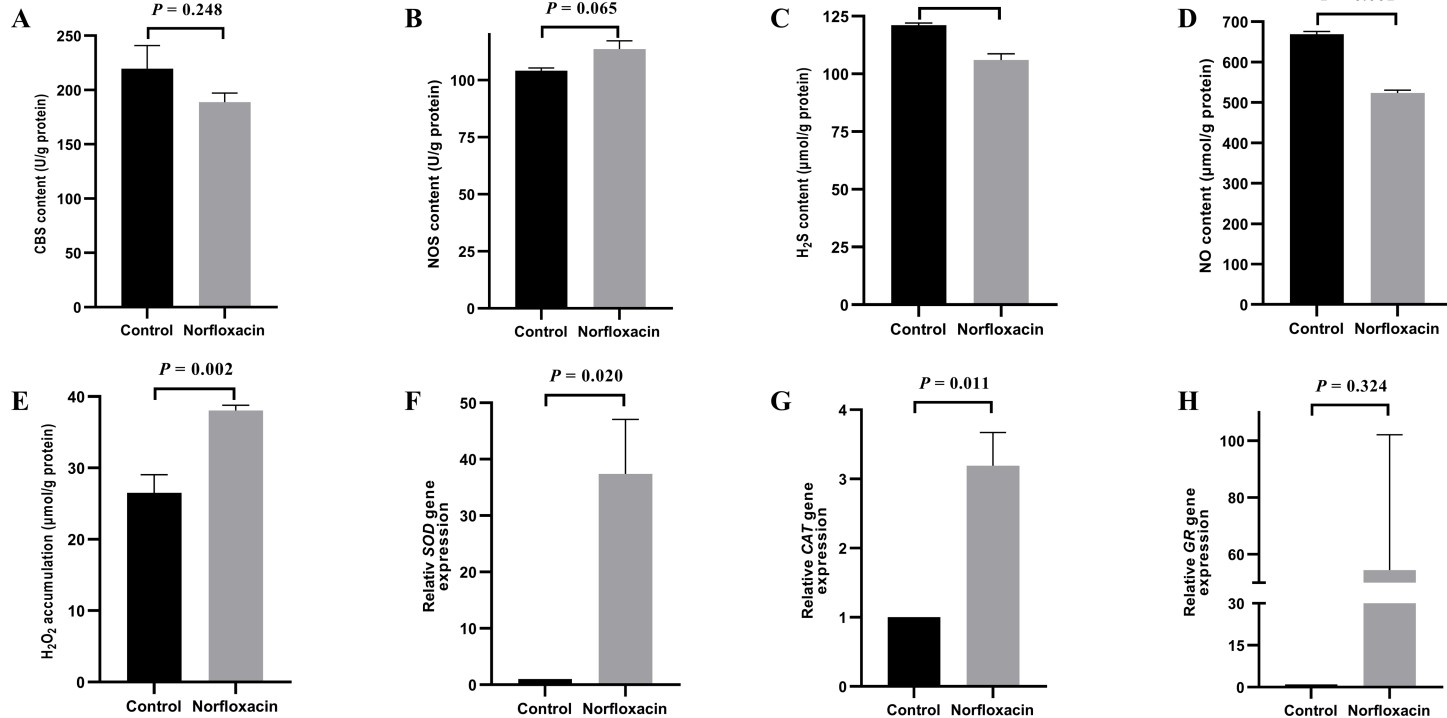

**Figure 1** **Effect of NOR on *Vibrio alginolyticus*.** (A) Effect of NOR on CBS content. (B) Effect of NOR-induced stress on the NOS content. (C) Effect of NOR-induced stress on the $H_2S$ content. (D) Effect of NOR-induced stress on NO content. (E) Effect of NOR-induced stress on $H_2O_2$ content. (F) Effect of NOR-induced stress on the relative expression of *SOD*. (G) Effect of NOR-induced stress on the relative expression of *CAT*. (H) Effect of NOR-induced stress on the relative expression of *GR*. The differences between the control and treatments were determined by unpaired t test of the statistical *t*-Test ($df = 4$), and F test was used to compare the variances. "*P*" stands for *P* value (two-tailed).

## Statistical analysis

Differences between the control and treatments were determined by the statistical *t*-Test method and F test with GraphPad Prism 5. Results were considered statistically significant if $P < 0.05$ and presented as means with standard error (SEM).

# RESULTS

## Minimum inhibitory concentration (MIC) of NOR

NOR minimum inhibitory concentration (MIC) against *V.alginolyticus* was 4 µg/mL. Hence subinhibitory concentration as 1/2 MIC concentration (2 µg/mL) of NOR was used to treat *V. alginolyticus* to assess the interaction of $H_2S$ and NO under NOR stress.

## Effect of NOR treatment on *V. alginolyticus*

Under 1/2 MIC NOR stress, CBS content decreased without a significant statistical difference ($P = 0.248$), and $H_2S$ content was lower than that in the control group ($P = 0.006$) (Figs. 1A and 1C). It is interesting that, NOR could evidently down-regulate NO content ($P < 0.001$) (Fig. 1D) though it had no significant influence on NOS content in cells ($P = 0.065$) (Fig. 1B). Besides, $H_2O_2$ accumulation ($P = 0.002$) and the relative expression level of *SOD* ($P = 0.020$) and *CAT* ($P = 0.011$) increased under NOR stress

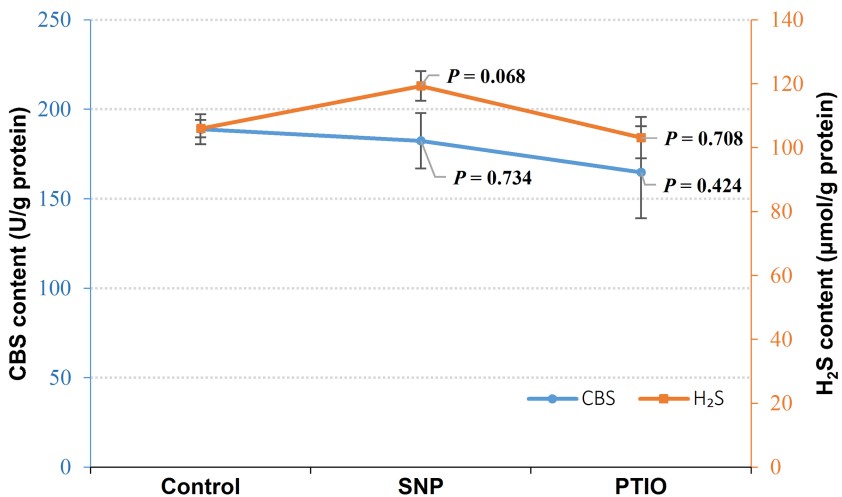

**Figure 2   CBS content and $H_2S$ content in *Vibrio alginolyticus* after treated with NO donor or scavenger under NOR-induced stress.** The differences between the control and treatments were determined by unpaired $t$ test of the statistical $t$-Test ($df = 4$), and F test was used to compare the variances. "$P$" stands for $P$ value (two-tailed).

compared to the control (Figs. 1E, 1F and 1G), and the relative expression level of *GR* also increased in cells treated with NOR ($P = 0.324$) (Fig. 1H).

### Effect of NO on CBS and $H_2S$ content in *V. alginolyticus*

Under NOR stress, the endogenous $H_2S$ level increased in the treatment groups with exogenous NO by SNP treatment ($P = 0.068$), whereas CBS content did not change evidently in the groups treated with SNP compared to the control (Fig. 2). Both of CBS ($P = 0.424$) and $H_2S$ ($P = 0.708$) content decreased in the group treated with NO scavenger of treatment (PTIO), though $P$-value failed to achieve a statistical significance (Fig. 2).

### Effect of $H_2S$ on NOS and NO content in *V. alginolyticus*

Exogenous $H_2S$ (NaSH treatment) improved NOS content in *V. alginolyticus* ($P = 0.348$, Fig. 3) and NOS content did not change significantly when treated with $H_2S$ scavenger (HT). Interestingly, NO content was up-regulated by both the exogenous $H_2S$ ($P = 0.189$) and $H_2S$ scavenger ($P = 0.129$) treatment compared to the control (Fig. 3).

### $H_2O_2$ accumulation and relative expression level of antioxidant-related genes under donors or scavengers

Under NOR stress, the relative expression level of *SOD* in *V. alginolyticus* treated with SNP decreased significantly ($P = 0.032$) compared to the control, but it increased slightly after treated with PTIO ($P = 0.191$) (Fig. 4A). Also, the relative expression level of *SOD* in the treatment group with NaSH was lower than that in the control, but it increased quickly when bacteria were treated with HT ($P = 0.168$) (Fig. 4A). It is noteworthy that, the relative expressions level of *CAT* decreased significantly in the treatment groups with different donors or scavengers (Data not shown).

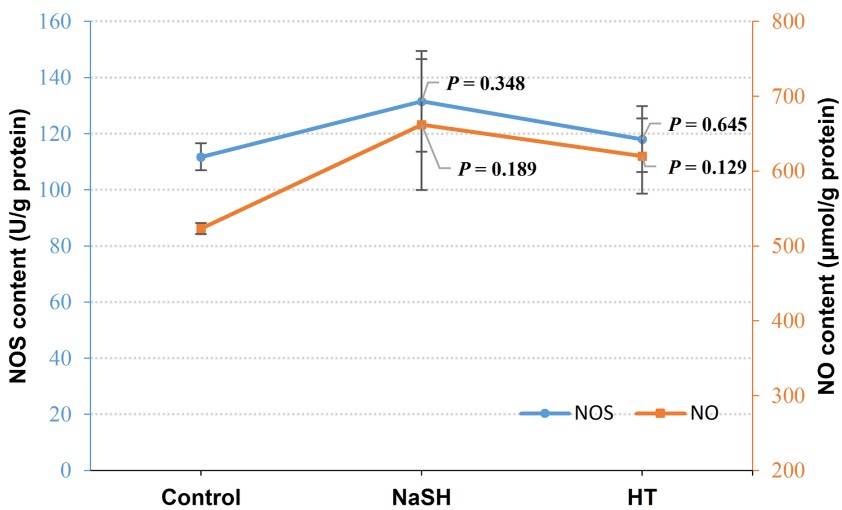

**Figure 3** **NOS content and NO content in *Vibrio alginolyticus* after treated with $H_2S$ donor or scavenger under NOR-induced stress.** The differences between the control and treatments were determined by unpaired $t$ test of the statistical $t$-Test ($df = 4$), and $F$ test was used to compare the variances. "$P$" stands for $P$ value (two-tailed).

Similar to the change of relative expression level of *SOD*, $H_2O_2$ accumulation in bacterial cells decreased significantly in the treatment group with SNP ($P = 0.003$), and also decreased in the treatment group with NaSH ($P = 0.759$), whereas it rose up when treated with PTIO ($P = 0.004$) or HT ($P < 0.001$) with significant difference (Fig. 4B) compared to the control group.

## DISCUSSION

Low dosage of ROS can act as a cellular signal in regulating the activation of protein kinases (*Tobiume et al., 2001*) and substance synthesis (*Steinite, Gailite & Ievinsh, 2004*) in a normal cell. Hence, low levels of ROS in cells are necessary and have a positive role on themselves. However, over-production of ROS induced by the environmental stress can influence on the balance of oxidant/reduction in cells, and will lead to serious destruction or dysfunction of cells (*Haigis & Yankner, 2010*; *Shukla, Mishra & Pant, 2011*). Many studies showed that $H_2S$ and NO play a great role in regulating the antioxidant system without an identical mechanism (*Chen et al., 2013*; *Li, 2013*). Interestingly, $H_2S$ has been proved to regulate the antioxidant system by enhancing the activities of the antioxidative enzymes (*Shatalin et al., 2011*; *Chen et al., 2013*), while NO was found to has a negative effect on the antioxidative enzymes (*Vital et al., 2008*), which suggest that it may act as a direct scavenger of ROS to decrease excessive $O_2^-$ and $H_2O_2$ (*Patel et al., 1999*; *O'Donnell & Freeman, 2001*). Though the effects of $H_2S$ and NO on the antioxidant system and the cross-talk between $H_2S$ and NO have already been studied in some species (*Li, 2013*), it still remains unknown in *V. alginolyticus*.

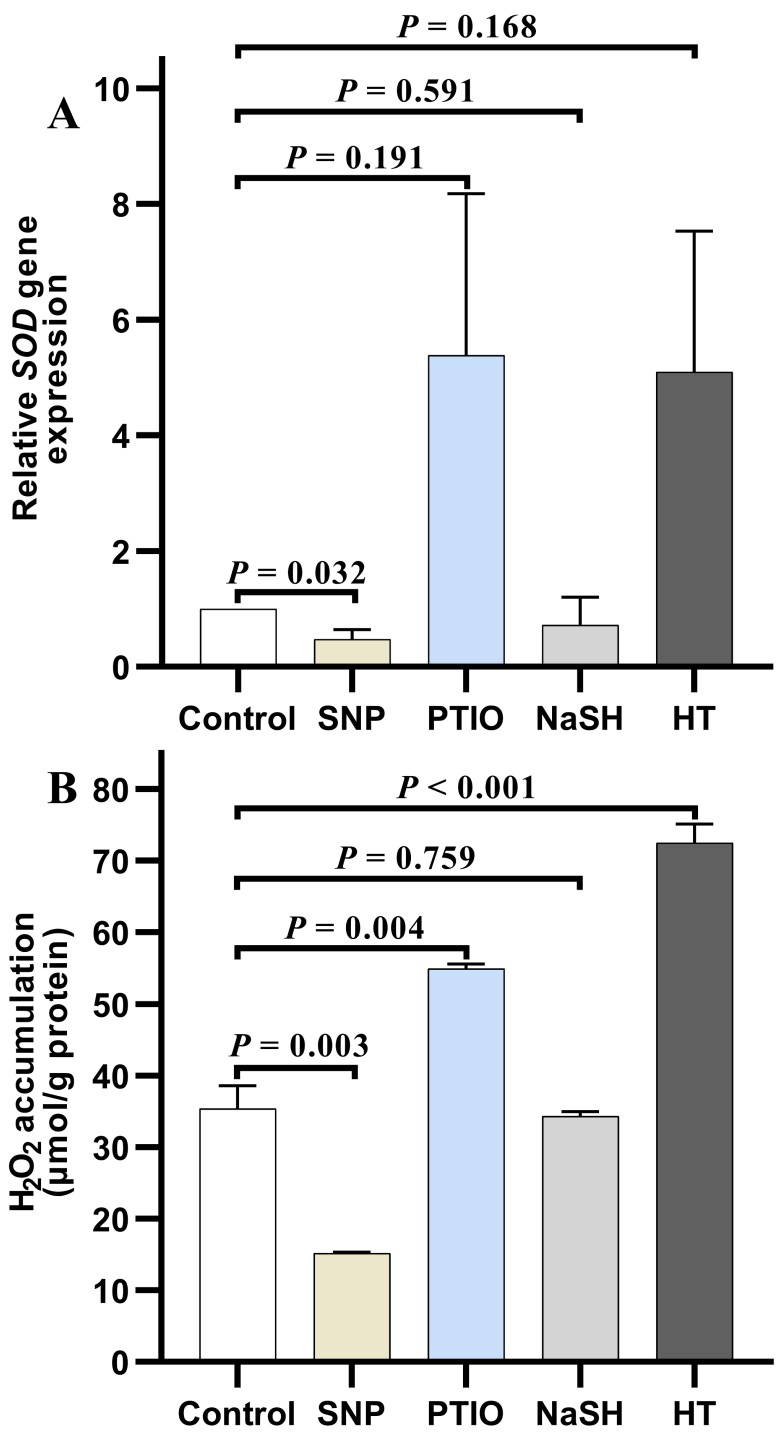

**Figure 4** **H₂O₂ accumulation and *SOD* relative expression level in *Vibrio alginolyticus* after treated with H₂S and NO and their donors or scavengers under NOR-induced stress.** (A) relative expression level of *SOD*. (B) H₂O₂ accumulation. The differences between the control and treatment were determined by unpaired t test of the statistical *t*-Test ($df = 4$), and $F$ test was used to compare the variances. "*P*" stands for *P* value (two-tailed).

In this work, it was initially found that $H_2S$ and NO contents as well as CBS content decreased significantly, while $H_2O_2$ accumulation and the relative expression of anti-oxidation related genes increased significantly in *V. alginolyticus* under NOR stress (Fig. 1). These showed that NOR could induce significantly $H_2O_2$ accumulation and the expression of anti-oxidation related genes in *V. alginolyticus*. Increased expression of antioxidant-related genes should be induced directly by accumulating $H_2O_2$ as a kind of ROS and a small molecular messenger (*Ding et al., 2012*). $H_2S$ content was positively related to the content of CBS. But we don't know whether the antibiotics of NOR affect the biosynthesis of NO, or the decrease in NO content were due to its consumption in alleviating the excessive ROS.

Under NOR stress, $H_2S$ content increased after treated with SNP (NO donor) while it decreased after treated with PTIO (NO scavenger) (Fig. 2), and NO content increased when treated with NaSH ($H_2S$ donor) and HT ($H_2S$ scavenger) (Fig. 3), indicating that there is a regulation and compensation mechanism between $H_2S$ and NO to response to NOR-induced stress, and $H_2S$ may be the upstream signal molecule in regulating the bio-synthesis of NO. However, the results presented that CBS content also decreased in the treatment of exogenous NO scavenger but no evident change after being treated with the exogenous NO treatment (Fig. 2), while NOS content was up-regulated by exogenous $H_2S$ but did not change significantly when $H_2S$ was scavenged (Fig. 3). These suggested that $H_2S$ and NO have a different effect on each other in regulating their synthases, but still demands more scientific details by the further studies to disclose it.

Many studies focused on the synergistical regulation between the two gases ($H_2S$ and NO) in smooth muscle, vasodilation in animal, and cobalt toxicity and heat tolerance in plant (*Hosoki, Matsuki & Kimura, 1997*; *Liew et al., 2007*; *Ozfidan-Konakci et al., 2020*; *Hassan, Maulood & Salihi, 2021*), whereas the research on the synergistic effect of the two gas molecules in regulating the antioxidant system of *V. alginolyticus* was hardly reported. In fact, the results mentioned above showed that there is significant correlation between the antioxidant system and two gas molecules of $H_2S$ and NO under the antibiotics stress. It showed that under NOR stress, the endogenous $H_2S$ and NO contents increased after treated respectively with exogenous NO and $H_2S$ (Figs. 2 and 3), while $H_2O_2$ accumulation and the relative expression level of *SOD* were down-regulated in *V. alginolyticus* (Fig. 4). That is to say, $H_2O_2$ induced by NOR was alleviated by the up-regulation of the exogenous NO and $H_2S$ on the anti-oxidative ability, while its accumulation was facilitated by the NO and $H_2S$ scavenger in the treatment with PTIO and HT, respectively (Fig. 4B). Decreased $H_2O_2$ down-regulated the gene expression level of anti-oxidative enzyme SOD. These also suggested that both NO and $H_2S$, directly or indirectly, take part in eliminating ROS induced by NOR in *V. alginolyticus*, and also showed that $H_2S$ and NO may work synergistically to regulate the antioxidant system in this bacterium under NOR stress. But interestingly, it had been found that NO content induced by abscisic acid (ABA) in plants decreased under the exogenous $H_2S$ (*Lisjak et al., 2011*). And NO content increased whereas $H_2S$ decreased in rats under chronic restraint stress (*Moustafa, 2021*), suggesting that there also exists antagonistic effect between these two gases in cells (*Decréau & Collman, 2015*). Therefore,

whether $H_2S$ and NO act synergistically on the antioxidant system is inconsistent in all the organisms and should be species-specific.

Finally, the medicines that can disturb the metabolic pathway of $H_2S$ and NO are potential to enhance the bactericidal effect, hence it is worthwhile to develop and synthesize these anti-vibrio drugs in the future, even though it needs more endeavors and cross studies in this fields. We believe that it was, and it will be a research hotspot in the future.

## CONCLUSION

Anti-oxidative ability in *V. alginolyticus* was respectively enhanced *via* the regulation of $H_2S$ and NO under NOR-induced stress, and there may be a crosstalk mechanism between $H_2S$ and NO to regulate the antioxidant system of *V. alginolyticus* treated with NOR, which lay a foundation for the research of regulation network of $H_2S$ and NO, and for study on the anti-drug mechanism. It also provides a novel target for the synthesis of anti-vibrio drugs.

### Funding

This work was supported by the grants from the Natural Science Foundation of Guangdong Province (No. 2014A030313604), the Graduate Education Innovation Program of Guangdong Ocean University (No. 201724) and the Foundation for the Distinguished Young Talents in Higher Education of Guangdong, China. The funders had no role in study design, data collection and analysis, decision to publish, or preparation of the manuscript.

### Grant Disclosures

The following grant information was disclosed by the authors:
Natural Science Foundation of Guangdong Province: 2014A030313604.
Graduate Education Innovation Program of Guangdong Ocean University: 201724.
Foundation for the Distinguished Young Talents in Higher Education of Guangdong, China.

### Competing Interests

The authors declare there are no competing interests.

### Author Contributions

- Shuhe Chen and Yu Ding conceived and designed the experiments, authored or reviewed drafts of the paper, and approved the final draft.
- Yunsheng Chang conceived and designed the experiments, performed the experiments, analyzed the data, prepared figures and/or tables, authored or reviewed drafts of the paper, and approved the final draft.
## Data Availability

The raw measurements and DNA sequence used in designing the primers are provided in the Supplementary Files.

## Supplemental Information

Supplemental information for this article can be found online at http://dx.doi.org/10.7717/peerj.12255#supplemental-information.

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

# PeerJ

**O'Donnell VB, Freeman BA. 2001.** Interactions between nitric oxide and lipid oxidation pathways: implications for vascular disease. *Circulation Research* **88**:12–21 DOI 10.1161/01.RES.88.1.12.

**Ozfidan-Konakci C, Yildiztugay E, Elbasan F, Kucukoduk M, Turkan I. 2020.** Hydrogen sulfide ($H_2S$) and nitric oxide (NO) alleviate cobalt toxicity in wheat (*Triticum aestivum* L.) by modulating photosynthesis, chloroplastic redox and antioxidant capacity. *Journal of Hazardous Materials* **388**:122061 DOI 10.1016/j.jhazmat.2020.122061.

**Patel RP, McAndrew HJ, Sellak CR, White H, Jo B. Freeman A, Darley-Usmar VM. 1999.** Biological aspects of reactive nitrogen species. *Biochimica Et Biophysica Acta/General Subjects* **1411**:385–400 DOI 10.1016/S0005-2728(99)00028-6.

**Rui H, Liu Q, Ma Y, Wang Q, Zhang Y. 2008.** Roles of LuxR in regulating extracellular alkaline serine protease A, extracellular polysaccharide and mobility of *Vibrio alginolyticus*. *FEMS Microbiology Letters* **285**:155–162 DOI 10.1111/j.1574-6968.2008.01185.

**Shatalin K, Shatalina E, Mironov A, Nudler E. 2011.** $H_2S$: a universal defense against antibiotics in bacteria. *Science* **334**:986–990 DOI 10.1126/science.1209855.

**Shukla V, Mishra SK, Pant HC. 2011.** Oxidative stress in neurodegeneration. *Advances in Pharmacological Sciences* **2011**:Article 572634 DOI 10.1155/2011/572634.

**Smith PK, Krohn R Il, Hermanson GT, Mallia AK, Gartner FH, Provenzano Md, Fujimoto EK, Goeke NM, Olson BJ, Klenk DC. 1985.** Measurement of protein using bicinchoninic acid. *Analytical Biochemistry* **150**:76–85 DOI 10.1016/0003-2697(85)90442-7.

**Steinite I, Gailite A, Ievinsh G. 2004.** Reactive oxygen and ethylene are involved in the regulation of regurgitant-induced responses in bean plants. *Journal of Plant Physiology* **161**:191–196 DOI 10.1078/0176-1617-01098.

**Stynen D, Goris A, Sarfati J, Latge JP. 1995.** A new sensitive sandwich enzyme-linked immunosorbent assay to detect galactofuran in patients with invasive aspergillosis. *Journal of Clinical Microbiology* **33**:497–500 DOI 10.1128/jcm.33.2.497-500.1995.

**Tobiume K, Matsuzawa A, Takahashi T, Nishitoh H, Morita K, Takeda K, Minowa O, Miyazono K, Noda T, Ichijo H. 2001.** ASK1 is required for sustained activations of JNK/p38 MAP kinases and apoptosis. *EMBO Reports* **2**:222–228 DOI 10.1093/embo-reports/kve046.

**Vital SA, Fowler RW, Virgen A, Gossett DR, Banks SW, Rodriguez J. 2008.** Opposing roles for superoxide and nitric oxide in the NaCl stress-induced upregulation of antioxidant enzyme activity in cotton callus tissue. *Environmental and Experimental Botany* **62**:60–68 DOI 10.1016/j.envexpbot.2007.07.006.

**Walsh C. 2000.** Molecular mechanisms that confer antibacterial drug resistance. *Nature* **406**:775–781 DOI 10.1038/35021219.

**Zheng D, Chen Z, Chen J, Zhuang X, Feng J, Li J. 2016.** Exogenous hydrogen sulfide exerts proliferation, anti-apoptosis, migration effects and accelerates cell cycle

progression in multiple myeloma cells via activating the Akt pathway Corrigendum. *Oncology Reports* **36**:1909–1916 DOI 10.3892/or.2021.7923.

**Zhou Z, Pang H, Ding Y, Cai J, Huang Y, Jian J, Wu Z. 2013.** VscO, a putative T3SS chaperone escort of *Vibrio alginolyticus*, contributes to virulence in fish and is a target for vaccine development. *Fish & Shellfish Immunology* **35**:1523–1531 DOI 10.1016/j.fsi.2013.08.017.