# Peer review of "Roles of H2S and NO in regulating the antioxidant system of Vibrio alginolyticus under norfloxacin stress"

_PeerJ, doi:10.7717/peerj.12255_

## Round 0.1 · original submission · Major Revisions

Dear Authors,

Your work has been assessed by three independent experts. Everyone agreed that this work could be published in PeerJ, but before that, it must be significantly corrected by you. Each of the Reviewers provided very detailed comments and remarks. They all noticed that the text itself needed language improvement.

Please read these comments and correct the manuscript accordingly.

·

Basic reporting

H2S and NO are important second messengers in organisms, which take part in many physiological processes. The manuscript (59645) reported that roles of H2S and NO in regulating the antioxidant system of vibrio alginolyticus under norfloxacin stress, which is import to understand the signaling role of H2S and NO. However, the following comments should be considered.
(1) The abbreviations should be given the full names at the first occurring place in the text.
(2) The H2S donor NaHS is a alkaline chemical, which might be affect the pH of the treatment solution, how to maintain the stability of pH.
(3) How to define the units of enzymes.
(4) Antioxidant enzymes usually are large family, which includes many members. In the text, the gene expression that is a member or a family for every enzyme is unclear.
(5) Figs.2, 3 should be labeled significance or no significance.
(6) Language should be carefully improved.

Experimental design

H2S and NO are important second messengers in organisms, which take part in many physiological processes. The manuscript (59645) reported that roles of H2S and NO in regulating the antioxidant system of vibrio alginolyticus under norfloxacin stress, which is import to understand the signaling role of H2S and NO. However, the following comments should be considered.
(1) The abbreviations should be given the full names at the first occurring place in the text.
(2) The H2S donor NaHS is a alkaline chemical, which might be affect the pH of the treatment solution, how to maintain the stability of pH.
(3) How to define the units of enzymes.
(4) Antioxidant enzymes usually are large family, which includes many members. In the text, the gene expression that is a member or a family for every enzyme is unclear.
(5) Figs.2, 3 should be labeled significance or no significance.
(6) Language should be carefully improved.

Validity of the findings

H2S and NO are important second messengers in organisms, which take part in many physiological processes. The manuscript (59645) reported that roles of H2S and NO in regulating the antioxidant system of vibrio alginolyticus under norfloxacin stress, which is import to understand the signaling role of H2S and NO. However, the following comments should be considered.
(1) The abbreviations should be given the full names at the first occurring place in the text.
(2) The H2S donor NaHS is a alkaline chemical, which might be affect the pH of the treatment solution, how to maintain the stability of pH.
(3) How to define the units of enzymes.
(4) Antioxidant enzymes usually are large family, which includes many members. In the text, the gene expression that is a member or a family for every enzyme is unclear.
(5) Figs.2, 3 should be labeled significance or no significance.
(6) Language should be carefully improved.

Additional comments

H2S and NO are important second messengers in organisms, which take part in many physiological processes. The manuscript (59645) reported that roles of H2S and NO in regulating the antioxidant system of vibrio alginolyticus under norfloxacin stress, which is import to understand the signaling role of H2S and NO. However, the following comments should be considered.
(1) The abbreviations should be given the full names at the first occurring place in the text.
(2) The H2S donor NaHS is a alkaline chemical, which might be affect the pH of the treatment solution, how to maintain the stability of pH.
(3) How to define the units of enzymes.
(4) Antioxidant enzymes usually are large family, which includes many members. In the text, the gene expression that is a member or a family for every enzyme is unclear.
(5) Figs.2, 3 should be labeled significance or no significance.
(6) Language should be carefully improved.

Reviewer 2 ·

Basic reporting

no comment

Experimental design

no comment

Validity of the findings

no comment

Additional comments

In this work, the authors evaluated the roles of H2S and NO in regulating the antioxidant system of Vibrio alginolyticus under norfloxacin stress. They revealed that anti-oxidative abilityin Vibrio alginolyticus was enhanced via the regulation of H2S and NO under norfloxacin (NOR)-induced stress, and H2S and NO may act synergistically to regulate the antioxidant system of Vibrio alginolyticus treated with NOR. The work is of interest of this journal readership and can be published.
And should address some comments as listed below:
1. The abstract section describes too much results, which should be reorganized the description structure.
2. The paper was poorly written and needs much editing to improve interpretation.
3. In the experimental part, the author did not cite any literature. What was the basis of the experimental operation?
4. The mechanism of H2S and NO on the antioxidant system of Vibrio alginolyticus should be illustrated.
5. The abstract should be rewritten to summarize the main research content.
6. To determine the interaction of H2S and NO, the author used SNP, PTIO and so on. How was the concentration and processing time determined? What was the influence of different solvent concentration and treatment time on the experimental results?
7. Please give the full name of the abbreviation for the first time use, such as MIC, SOD.
8. The following related papers on this topic maybe referenced for possible improving this manuscript:
Proteomic analysis reveals the protective role of exogenous hydrogen sulfide …., NITRIC OXIDE-BIOLOGY AND CHEMISTRY Volume: ‏ 111 Pages: ‏ 14-30 Published: ‏ JUN 1 2021
The vasodilatory mechanism of nitric oxide and hydrogen sulfide ….., EXPERIMENTAL AND THERAPEUTIC MEDICINE Volume: ‏ 21 Issue: ‏ 3 Article Number: 214 Published: ‏ MAR 2021
Imaging of anti-inflammatory effects of HNO via a near-infrared fluorescent probe ….., JOURNAL OF MATERIALS CHEMISTRY B Volume: ‏ 7 Issue: ‏ 2 Pages: ‏ 305-313 Published: ‏ JAN 14 2019
A highly selective turn-on near-infrared fluorescent probe for hydrogen sulfide ….., CHEMICAL COMMUNICATIONS Volume: ‏ 48 Issue: ‏ 96 Pages: ‏ 11757-11759 Published: ‏2012
Changes in nitric oxide, carbon monoxide, hydrogen sulfide ….., FREE RADICAL BIOLOGY AND MEDICINE Volume: ‏ 162 Pages: ‏ 353-366 Published: ‏ JAN 2021
Nitric Oxide and Hydrogen Sulfide Coordinately Reduce ……, ANTIOXIDANTS Volume: ‏ 10 Issue: ‏ 1 Article Number: 108 Published: ‏ JAN 2021
Nitric oxide and hydrogen sulfide protect plasma ……, PLANT PHYSIOLOGY AND BIOCHEMISTRY Volume: ‏ 157 Pages: ‏ 244-255 Published: ‏ DEC 2020
Hydrogen sulfide (H2S) and nitric oxide (NO) alleviate cobalt toxicity ……, JOURNAL OF HAZARDOUS MATERIALS Volume: ‏ 388 Article Number: 122061 Published: ‏ APR 15 2020

Reviewer 3 ·

Basic reporting

All the comments are provided in comments to author section

Experimental design

All the comments are provided in comments to author section

Validity of the findings

All the comments are provided in comments to author section

Additional comments

I have now finished the review on the manuscript titled " Roles of H2S and NO in regulating the antioxidant system of Vibrio alginolyticus under norfloxacin stress”. The problem of antibiotics stress has worsened over the last decade particularly in the developing countries. Besides, the intensity to tolerate varying degrees of antibiotics stress is significantly different among different microbial and plant species. Therefore, there is a need to evaluate role of antibiotics and their tolerance potential among different organisms. The authors studied the interactive effect of H2S and NO in regulating antibiotics stress and it contains valuable information. However, the following comments may be incorporated to improve the quality of the publication.
There are few grammatical errors in sentence formation as well as spelling errors. I am not mentioning the line numbers here and suggest a through reading of the manuscript.
Materials and Methods part is very important and should be reproducible. I have several concerns regarding different protocols and execution of different work. My main points are as under;
How did authors dissolve antibiotics?
The authors did not provide the references of protocols they followed in a number of places in materials and methods.
The authors have several typographic errors throughout the manuscript. They are suggested to read the manuscript carefully for typographic errors and grammatical mistakes.
The results are presented nicely. However, the authors have big standard errors. Is there some problem with making graphs?
The discussion part should include logical reasoning of the results obtained rather than just comparing the results with the literature cited.
The conclusion part is weak. The authors are suggested to improve it and also include the contribution of the results obtained to the existing knowledge. The authors are also encouraged to include how researchers could get benefit and plan their future research based on this experiment.
This paper contains lot of data and information that could be useful for the readers. Therefore, I suggest a minor revision to improve the scientific quality of the manuscript.

---

## Round 0.2 · accepted · Accept

I am very pleased to inform you that the reviewer has accepted all the corrections you have made and that there are no more comments. In connection with the above, the work can be published.
With best regards,

Reviewer 2 ·

Basic reporting

OK

Experimental design

OK

Validity of the findings

OK

Additional comments

The authors have satisfactorily responded to all the questions and made the necessary changes to the manuscript. I have no further questions and suggest the acceptance of the revised manuscript.